# Guava Detection and Pose Estimation Using a Low-Cost RGB-D Sensor in the Field

**DOI:** 10.3390/s19020428

**Published:** 2019-01-21

**Authors:** Guichao Lin, Yunchao Tang, Xiangjun Zou, Juntao Xiong, Jinhui Li

**Affiliations:** 1Key Laboratory of Key Technology on Agricultural Machine and Equipment, Ministry of Education, South China Agricultural University, Guangzhou 510642, China; guichaolin@126.com (G.L.); xiongjt@scau.edu.cn (J.X.); lijinhui29@163.com (J.L.); 2College of Mechanical and Automotive Engineering, Chuzhou University, Chuzhou 239000, China; 3School of Urban and Rural Construction, Zhongkai University of Agriculture and Engineering, Guangzhou 510006, China

**Keywords:** guava detection, pose estimation, fully convolutional network, branch reconstruction, RGB-D sensor

## Abstract

Fruit detection in real outdoor conditions is necessary for automatic guava harvesting, and the branch-dependent pose of fruits is also crucial to guide a robot to approach and detach the target fruit without colliding with its mother branch. To conduct automatic, collision-free picking, this study investigates a fruit detection and pose estimation method by using a low-cost red–green–blue–depth (RGB-D) sensor. A state-of-the-art fully convolutional network is first deployed to segment the RGB image to output a fruit and branch binary map. Based on the fruit binary map and RGB-D depth image, Euclidean clustering is then applied to group the point cloud into a set of individual fruits. Next, a multiple three-dimensional (3D) line-segments detection method is developed to reconstruct the segmented branches. Finally, the 3D pose of the fruit is estimated using its center position and nearest branch information. A dataset was acquired in an outdoor orchard to evaluate the performance of the proposed method. Quantitative experiments showed that the precision and recall of guava fruit detection were 0.983 and 0.948, respectively; the 3D pose error was 23.43° ± 14.18°; and the execution time per fruit was 0.565 s. The results demonstrate that the developed method can be applied to a guava-harvesting robot.

## 1. Introduction

Guava harvesting is labor-intensive, time-consuming, and costly work. The aging population and growing urbanization in China have resulted in an older agricultural labor force [1], which is becoming a potential threat to fruit harvesting. Therefore, it is urgent to develop an automatic guava-harvesting robot that can work in the field. In-field fruit detection is an important aspect of a harvesting robot [2], containing many challenges including varying illuminations, occlusion caused by leaves and branches, and color variations in fruit. Additionally, if only the fruit position information is available, the end-effector of the harvesting robot is likely to have collisions with the mother branch of the fruit when moving toward a fruit, hence lowering the harvest success rate. Thus, for each fruit, estimating a three-dimensional (3D) pose relative to its mother branch along which the end-effector can approach the fruit without colliding with the branch is very important. In this work, the fruit pose is defined as a vector that passes through the fruit center and is perpendicular to the mother branch of the fruit. Bac et al. has shown that such a pose could increase the grasp success rate from 41% to 61% [3]. Guava fruit detection and pose estimation were investigated in this study.

Fruit detection has been extensively studied. Analysis of red–green–blue (RGB) images captured from charge-coupled device (CCD) cameras is a common approach [4]. For instance, Song et al. [5] used a high-resolution CCD camera to acquire images from which a naïve Bayes classifier and support vector machine (SVM) trained on color and texture features were applied to recognize peppers. Sengupta and Lee [6] utilized a circular Hough transform (CHT) and texture-based SVM to detect citrus. Qureshi et al. [7] used a color camera at night with a shape-based detector to locate mangoes. Our research group studied a color-based [8] and texture-based [9] AdaBoost classifier to detect fruits. In these methods, convolutional neural networks (CNN) have shown impressive performance. For instance, Sa et al. [10] presented a multi-modal faster region-based CNN (Faster R-CNN) to detect peppers from near-infrared (NIR) and RGB images captured from a multispectral camera, and achieved an F1 score of 0.838. Stein et al. [11] developed a multi-sensor-based method that used a Faster R-CNN, multiple RGB images, and light detection and ranging (LiDAR) data to predict apple yield with an error rate of 1.36% per tree. Bargoti and Underwood [12] also deployed a Faster R-CNN to detect apples and mangoes in outdoor orchards, and realized an F1-score larger than 0.9. In another study, Bargoti and Underwood [13] developed a shallow CNN to segment RGB images followed by watershed segmentation and CHT to detect apples with an F1 score of 0.858. As red–green–blue–depth (RGB-D) images encode the color and 3D geometry of the object, and the RGB-D depth image is invariant to illumination changes, RGB-D images are more informative than RGB images. Therefore, there has been an increase in using RGB-D sensors to detect fruits [4]. Barnea et al. [14] combined image highlights, 3D normal features, and fruit symmetry planes to detect sweet peppers from RGB-D images generated by a depth and an RGB camera, and realized a mean average precision of 0.52. Nguyen et al. [15] used a color and depth filter, Euclidean clustering [16], and CHT to detect apples on trees. The algorithm can detect 100% of visible apples and 82% of occluded apples by using a low-cost RGB-D sensor. Kusumam et al. [4] first performed depth filtering and outlier removal to exclude useless points, and then used Euclidean clustering to group point clouds into a set of clusters, and finally applied an SVM classifier to remove false positives to achieve broccoli head detection. A detection accuracy of 0.947 was obtained. Wang et al. [17] first utilized a cascade classifier to detect mangoes, and then used an ellipse fitting method and the RGB-D depth image to estimate the mango size. A root mean square error of 4.9 mm was reported. In conclusion, both the CNN-based and RGB-D-based methods show promising results on fruit detection in the fields. This study fuses these two methods to detect guava fruits in outdoor conditions.

Currently, there is a small amount of research on fruit pose estimation. Depth-sensing cameras are used typically. Eizentals and Oka [18] first used a LiDAR sensor to acquire pepper point clouds, and then applied a coherent point drift algorithm to align the point clouds with a given 3D model to compute the affine transformations that represent the fruit poses. Pose errors of 77.6% of the fruits were within 25 mm in indoor conditions, although the algorithm was inefficient (14.50 s per fruit). Lehnert et al. [19] employed a multivariate Gaussian model to detect pepper fruits from RGB-D images, and then used a nonlinear least squares method to fit a superellipsoid to the detected pepper point cloud to estimate its pose. Li et al. [20] used an Intel RealSense depth sensor to acquire point clouds, and then estimated the poses of sweet peppers by detecting their symmetry axes. The average error was about $7.4{}^{\circ }$. These methods do not take into account the relative position of the fruit to its mother branch, probably resulting in a collision between the robot and branch when applied to a guava-harvesting robot.

Branch information is crucial for estimating the pose of the guava fruit relative to its mother branch. Branch or stem detection has been widely studied. Van Henten et al. [21] took the advantage of the high reflectance of plant stems in NIR images to detect cucumber stems. Lu et al. [22] used multispectral imaging technology to recognize citrus branches. Noble and Li [23] analyzed NIR images at varying spectral bands from a spectrophotometer to segment cucumber plant parts, and reported that reliable lighting conditions were required to obtain promising results. Bac et al. [24] used a hyperspectral camera to acquire images in which a classification and regression trees classifier (CART) is deployed to segment pepper plant parts as soft or hard obstacles, and showed that the result was insufficient for planning collision-free trajectories. In another study, Bac et al. [25] developed a vision system that used support wires twisted around the stems as visual cues to detect pepper stems. This method completed a true positive rate of 0.94, but the usage of support wires limits its application. Zhang and Xu [26] proposed an unsupervised conditional random field algorithm to cluster tomato plants into fruits, leaves, and stems, which obtained a high accuracy, but took 94.45 s per image. Recently, Majeed et al. [27] applied a CNN, SegNet [28], for segmenting apple tree trunks and branches from RGB-D images, and obtained promising results. Therefore, the CNN could be used to segment guava branches.

The objective of this study was to develop a vision sensing algorithm to detect the guava fruit and estimate its pose in real outdoor conditions using a low-cost RGB-D sensor. The pipelines of the study (i) employed a fully convolutional network (FCN) [27,28,29] to segment guava fruits and branches simultaneously from RGB images, (ii) used Euclidean clustering [4,15,16] to detect all the 3D fruits from the fruit binary map output by a FCN, (iii) established a 3D line segments detection method to reconstruct the branches from the branch binary map, and (iv) estimated the 3D pose of the fruit using its center position and mother branch information [3].

## 2. Materials and Methods

### 2.1. Vision Sensing System

The guava-harvesting robot and its vision sensing system are shown in Figure 1. The vision sensing system comprised a low-cost RGB-D sensor and a sensing algorithm. The RGB-D sensor that was used was the Kinect V2 made by Microsoft Inc., which consists of an infrared (IR) light source, an IR camera, and an RGB camera. The IR light source actively illuminates the object using modulated light, and the IR camera can detect the phase shift of the received light to measure a sensor-to-target distance and result in a depth image of 424 × 512 pixels. The RGB camera can create an RGB image of 1920 × 1080 pixels. The RGB and depth images need to be aligned before application, because their resolutions are different. The MapDepthFrameToColorSpace function from the Kinect for Windows SDK 2.0 was used to implement the image alignment operation. In this way, the RGB image is resized to 424 × 512 pixels to match the depth image. The depth data can be converted to 3D coordinates by the following equation [2]:(1)$$\{\begin{array}{c} z_{i}=I_{depth}\left(u_{i},v_{i}\right)\\ x_{i}=z_{i}\left(u_{i}-U_{x}\right)/f_{x}\\ y_{i}=z_{i}\left(v_{i}-U_{y}\right)/f_{y} \end{array}$$
where $\left(x_{i},y_{i},z_{i}\right)$ are the 3D coordinates of pixel *i*; $\left(u_{i},v_{i}\right)$ are the pixel coordinates of pixel *i*; $I_{depth}$ is the depth image; $\left(U_{x},U_{y}\right)$ are the pixel coordinates of the principal point of the IR camera; and $\left(f_{x},\;f_{y}\right)$ are the focal lengths of the IR camera. $U_{x}$, $U_{y}$, $f_{x}$, and $f_{y}$ were estimated using the calibration method developed by Zhang [30]. In the experiment, the minimum distance from the Kinect V2 to the guava tree was set to 550 mm.

### 2.2. Fruit Detection and Pose Estimation Algorithm

The pipeline of the developed vision sensing algorithm is shown in Figure 2. It processes the RGB and depth images and comprises the following functions: (i) using an FCN model to segment guava fruits and branches, (ii) applying Euclidean clustering to obtain all of the individual fruits from fruit point clouds, (iii) presenting a multiple 3D line segments detection method to reconstruct the branches from branch point clouds, and (iv) estimating the pose of the fruit relative to its mother branch. Note that each point cloud is created from a single viewpoint, so it only contains part of the geometry of the object. Nevertheless, partial point clouds are sufficient for fruit detection [15].

#### 2.2.1. Image Segmentation

The objective of image segmentation is to segment the guava fruits and branches from aligned RGB images. A state-of-the-art FCN model [29] is used. It rewrites the fully connected layers of VGG-16 [31] or GoogLeNet [32] into the fully convolutional layers, thus outputting a dense prediction map. The first row in Figure 3 shows a VGG-16-based FCN model, which is made up of a succession of convolutional, max-pooling, and deconvolutional layers. The deconvolutional layer is used to linearly up-sample the coarse maps to dense maps. As a total of five max-pooling operations are performed by FCN, the size of the output of the conv7 layer is 1/32 of the input image. Consequently, a deconvolution operation with a stride of 32 will reduce the detailed information of the final prediction. To deal with this problem, a skip strategy [29] is presented to fuse the outputs from the conv7 layer and some lower layers (second row in Figure 3) to refine the final prediction. By combining the coarse layer and fine layer, the new model considers not only global structures, but also local details, thus improving the segmentation accuracy. More details can be found in [29].

Here, the input to FCN is the aligned RGB image with a resolution of 424 × 512 pixels. A fine-tuning strategy is used to train the FCN model on our small-sized training set (Section 3.1). The details of fine-tuning include (i) using bilinear interpolation [29] to initialize the parameters of all the deconvolutional layers, (ii) initializing other parameters by simply inheriting the parameters of a publicly available FCN model [33], and (iii) using an Adam solver [34] with a learning rate of 0.0001 to update the FCN parameters. In addition, 10-fold cross-validation was applied over the training set to determine an optimal FCN. After training, FCN can be applied to segment the aligned RGB image to output a fruit and a branch binary map. A visual example is shown in Figure 4. 

#### 2.2.2. Fruit Detection and Localization

Since the FCN is unable to identify how many fruits are in the fruit binary map that it outputs and may segment adjacent fruits into a single region, it is necessary to extract all of the individual fruits from the FCN output in order to realize fruit detection.

Let $p_{fruit}$ denote a set of pixels that belong to the fruit class in the fruit binary map outputted by FCN. $p_{fruit}$ can be transformed into a point cloud (Figure 5a), which is defined as $P_{fruit}$, by using Equation (1). Euclidean clustering [16] is performed to group $P_{fruit}$ into a set of clusters, with each cluster representing a single fruit. Euclidean clustering is a region growing-based clustering method. It first selects a point from $P_{fruit}$ as an initial cluster, and then grows this cluster by searching the nearest neighbors of each point in the cluster within a given threshold. Once this cluster stops growing, another unclustered point is selected as the initial value of the new cluster, which is enlarged in the same way. When no unclustered point exists, a certain number of clusters will be obtained. A clustering example is shown in Figure 5b.

It is worthwhile to note that determining a proper value for the threshold is very important. A large threshold will result in merging adjacent fruits as a single fruit, while a small one will split a fruit into several parts. In experiments, this threshold was set to 4 mm, as suggested by [4]. Additionally, for each point in $P_{fruit}$, finding its nearest neighbors within a threshold requires $\left|P_{fruit}\right|$ calculations, which is very time-consuming. A *K*d-tree algorithm is used to optimize the computation. 

The next task is to determine the fruit center position. Two methods based on (i) the bounding box [4] and (ii) sphere fitting are presented. The bounding box method uses the mean of a fruit cluster as the fruit center position. The sphere fitting method applies linear least squares fitting to fit a sphere model for a fruit cluster, and uses the sphere center as the fruit center position. The former method is computationally efficient, while the latter is relatively precise in positioning.

#### 2.2.3. Branch Reconstruction

In this paper, the fruit pose is defined as the orientation of the fruit center relative to its mother branch. To achieve reliable 3D pose estimation, it is crucial to reconstruct the branches. 

A skeleton extraction algorithm [35] is first performed to thin the branches from the FCN output to decrease the scale of the branch point cloud to improve the computational efficiency. Let *A* denote a binary branch map generated by FCN, and *S* denote the branch skeletons. *S* can be computed by:(2)$$S=\overset{K}{\underset{k=0}{\cup }}\left(A⊖kB\right)-\left(A⊖kB\right){}^{\circ }B$$
where *B* is a structure element; (A⊖kB) means performing *k* morphological erosion operations on *A* using the structure element *B*; ${}^{\circ }$ refers to the morphological opening operation; and K=max{k|(A⊖kB)≠∅}. Figure 6a shows an example of branch skeleton extraction.

Figure 6b shows the point cloud of the branch skeletons. As can be seen, each branch is in fact a complicated 3D curve that is hard to fit. Inspired by Botterill et al. [36], straight line segments were used to approximate the branches. A random sample consensus algorithm (RANSAC)-based 3D line-segments detection method is investigated in this study, which comprises the following steps: Step 1. Randomly select two points, *p*_1_ and *p*_2_, from the branch point cloud $P_{branch}$ to calculate the parameters of a line candidate as $\left(p_{1},t_{1}\right)$ where $t_{1}=\left(p_{1}-p_{2}\right)/\left|p_{1}-p_{2}\right|$, then search inliers that fit this line within a threshold. The threshold was set to 15 mm in our experiments.Step 2. Repeat Step 1 *N* times (*N* was set to 4000 in experiments). If the number of inliers of the line model with the largest number of inliers is larger than a predefined threshold (which was set to 40 in the experiments), choose this line model as a line segment, subtract the inliers from $P_{branch}$, and go to Step 3. Otherwise, stop the line detection.Step 3. Repeat Step 2 until the $P_{branch}$ is empty. 

The proposed line detection method can detect a set of line segments. An example is shown in Figure 6c. It is worthwhile to note that Botterill et al. [36] used a 2D line-segments detection method, which may wrongly detect two line segments that are collinear in the 2D plane, but non-collinear in the 3D space as one line segment. The developed 3D line-segments detection method can address this problem.

#### 2.2.4. Fruit Pose Estimation

The final step of the proposed pipeline is to estimate the fruit pose so that the robot can approach the fruit along its pose for collision-free picking. Firstly, for an arbitrary detected fruit, such as the *i*th fruit, its mother branch is determined by finding the nearest line segment by:(3)$$j=\underset{k\in \left[1,\ldots ,M\right]}{\mathrm{argmin}}\left|\left(c_{i}-p_{k}\right)-{\left(c_{i}-p_{k}\right)}^{T}t_{k}t_{k}\right|$$
where $c_{i}$ is the center position of the *i*th fruit; $\left(p_{k},t_{k}\right)$ refer to the parameters of the *k*th line segment, where $p_{k}$ is a point that the line passes through, and $t_{k}$ is the unit direction vector of the line; and *M* is the total number of line segments detected. Then, the nearest point of the mother branch to the fruit center is calculated by $n_{i}=p_{j}+{\left(c_{i}-p_{j}\right)}^{T}t_{j}t_{j}$ (Figure 7a). Finally, the fruit pose is estimated by ${\tilde{q}}_{i}=\left(c_{i}-n_{i}\right)/\left|c_{i}-n_{i}\right|$. A pose estimation example is shown in Figure 7b.

## 3. Datasets

### 3.1. Image Acquisition

The setup as mentioned in Section 2.1 was used to acquire RGB-D images. The experimental site was located in a commercial guava orchard in Guangzhou, China. The guava variety is *carmine*. The image collection time was from 12:00–17:00 on 8 July 2018, 08:00–17:00 on 15 September 2018, and 08:00–17:00 on 13 October 2018. No artificial light source was used; i.e., the collected images contained all kinds of illuminations. There were 437 RGB-D images captured in total. To train and test the proposed algorithm, the image dataset was divided into a training set and a test set. The training set contained approximately 80% of the RGB-D images in the dataset, and the test set included the remaining images, as suggested by [10].

### 3.2. Ground Truth

To train the FCN and evaluate its performance, all of the images in the training set and test set needed to be annotated. The Image Labeler app in MATLAB was used to manually label each pixel in the images as a background, fruit, or branch class (Figure 8a). The annotation task was very time-consuming, and took five days.

To evaluate the precision of the fruit pose estimation, the ground-truth pose of each fruit in the test set should be measured. The following steps were adopted to measure the ground-truth pose: (i) using MATLAB’s Image Labeler app to manually label each fruit and its mother branch in the aligned RGB image (Figure 8b); (ii) using a robust RANSAC-based sphere fitting method [37] to fit the fruit point cloud to determine the fruit center position; (iii) determining the nearest point of the mother branch point cloud to the fruit center; and (iv) using the vector that passes through this nearest point and points to the fruit center as the fruit pose. Note that if the mother branch of a fruit in the test set was invisible, the ground-truth pose of this fruit was not measured. In total, the ground-truth poses of 63.55% of the fruits in the test set were measured.

## 4. Results and Discussions

To evaluate the performance of the proposed vision sensing algorithm, quantitative experiments were carried out. All of the codes were programmed in MATLAB 2017b, except for the FCN, which was implemented in Caffe [38] using a publicly available code [33].

### 4.1. Image Segmentation Experiment

Image segmentation performance is crucial for the proposed pipeline. The mean accuracy and mean region intersection over union (IOU) were used to evaluate the segmentation performance of the FCN, which takes the following form:(4)$$\mathrm{mean}\;\mathrm{accuracy}=n_{ii}/t_{i}$$
(5)$$\mathrm{IOU}=n_{ii}/\left(t_{i}+\sum_{j=1}^{n_{cl}}n_{ji}-n_{ii}\right)$$
where $n_{cl}$ is the number of classes and equals three in our case; $n_{ij}$ is the number of pixels that belong to class *i*, but are predicted for class *j*; and $t_{i}$ is the number of pixels that belong to class *i*, and equals $\sum_{j=1}^{n_{cl}}n_{ij}$. SegNet [28] and CART were used as baseline algorithms. SegNet is a CNN for image segmentation; it was trained on the training set, as mentioned in Section 3.1. CART was employed by Bac et al. [24] to model pixel-wise features to segment plant parts; here, it was trained on RGB and Gabor-like texture features [39] extracted from our training set. 

Table 1 showed the segmentation results of FCN, SegNet, and CART on our test set. Obviously, the FCN and SegNet outperformed CART, because CNN-based algorithms have a great number of filter banks that can extract a wide context for each pixel well, which enhances the segmentation performance, whereas CART only uses several hand-engineered filter banks. Additionally, the FCN performed better than SegNet, mainly because an over-fitting problem occurred when training the SegNet. In short, the FCN is more suitable for small-sized datasets, and hence could be applied to segment guava fruits and branches.

It is worthy of note that in the experiment, some guava branches were easily classified as the background by the FCN, resulting in a lower mean accuracy and IOU than the fruits. Future work could fuse multi-source images to improve the vision saliency of the branches and fuse the outputs of more lower layers of the FCN, improving the performance of the branch segmentation.

### 4.2. Fruit Detection Experiment

To evaluate the guava detection performance, precision and recall were used. Precision is the ratio of the number of true positives to the number of detections; recall is the ratio of the number of true positives to the number of fruits in the images. A state-of-the-art RGB-D-based detection method presented by Kusumam et al. [4] was used as the comparison algorithm. It consisted of the following steps: (i) depth filtering, (ii) Euclidean clustering, and (iii) point cloud classification via an SVM classifier trained on angular features.

Table 2 showed the detection results over the test set. The proposed algorithm realized a precision of 0.983 and a recall of 0.949, both of which were better than the comparison algorithm. In [4], Euclidean clustering was prone to group fruits and adjacent leaves as single objects, decreasing the detection performance. It can be concluded that the proposed algorithm is robust for detecting guava fruits in the field, and could be applicable to a guava-harvesting robot.

The detection results are shown in Figure 10 (Section 4.3). Some unsuccessful detections of the proposed algorithm are shown in Figure 9. There were two main reasons for the lack of success: (i) in strong sunlight, the standard deviation of the depth data of Kinect V2 for a sensor-to-target distance z = 1 m was up to 32 mm as reported by Fankhauser et al. [40], which would increase the invalid points in the point cloud and result in detection failures (Figure 9a); and (ii) on four occasions in experiments, Euclidean clustering clustered two neighboring fruits as a single fruit (Figure 9b). To address these two problems, the following solutions can be adopted: (i) avoiding using Kinect V2 at noon or using a light shield to block sunlight [15]; and (ii) adding curvature cues in Euclidean clustering, because the curvatures between two neighboring fruits are obviously larger than the fruit surfaces.

### 4.3. Pose Estimation Experiment

The 3D pose estimation experiment was carried out over the test set. The fruit pose error was measured by calculating the angle between the estimated and the ground-truth pose:(6)$$\theta_{i}=\mathrm{acos}\left(\frac{{\tilde{q}}_{i}{}^{T}q_{i}}{\left|{\tilde{q}}_{i}\right|\left|q_{i}\right|}\right)$$
where ${\tilde{q}}_{i}$ and $q_{i}$ are the estimated and ground-truth poses of the *i*th fruit, respectively. Smaller angles correspond to smaller errors, and higher angles correspond to higher errors. Since the mean error and standard deviation may be affected by large errors, the median error (MEDE) and median absolute deviation (MAD) were used to analyze the pose errors, which take the following form [4]:(7)$$\{\begin{array}{c} \mathrm{MEDE}=median\left\{\theta_{i}\right\}\\ \mathrm{MAD}=median\left\{\left|\theta_{i}-\mathrm{MEDE}\right|\right\} \end{array}$$
where *median*() is the median operation.

Table 3 shows the experimental results of our two algorithms for the estimation of fruit poses. The 3D pose error of the bounding box-based method was $25.41{}^{\circ }\pm 14.73{}^{\circ }$, while the sphere fitting-based method was $23.43{}^{\circ }\pm 14.18{}^{\circ }$. The frequency of the 3D pose error within certain angle limits was also analyzed. The results for limits of $45{}^{\circ }$, $35{}^{\circ }$, and $25{}^{\circ }$ can be found in Table 4. These results suggested that the sphere fitting method was more suitable for pose estimation. However, in terms of the computational efficiency, the bounding box-based method was more attractive.

In practical applications, the errors of the fruit poses under a certain limit would not cause the robot to collide with the branch. Further study will conduct field experiments to determine such limits and the contribution of the fruit pose information to the harvest success rate. It is important to note that for branches without fruits, if they block the end-effector from approaching the fruit, the path planning algorithm proposed by our research group [41] can be utilized to avoid them.

The visual results of fruit pose estimation are shown in Figure 10. Some failures are shown in Figure 11. The causes of failures were grouped into two categories: (i) when the mother branch of a fruit was very thin or invisible, the FCN was unlikely to recognize it, and hence the pose of this fruit relative to its mother branch could not be estimated (Figure 11a); and (ii) the pose estimation algorithm used the nearest branch as the mother branch of a fruit, which may be a false mother branch (Figure 11b). To deal with these problems, the following solutions can be considered: (i) cultivating a new variety with fewer leaves, which would be more suitable for robotic harvesting [42] or improving the FCN performance; and (ii) utilizing the prior knowledge of the fruit; for example, considering that the mother branch tends to be above the fruit.

### 4.4. Time Efficiency Analysis

The vision sensing system should not slow the robotic system during harvesting to ensure the production, and so its real-time performance was analyzed. The real-time experiment was implemented on a computer running a 64-bit Windows 10 system with 16 GB RAM, NVIDIA GeForce GTX 1060 6GB GPU, and Intel core i5-3210M CPU. Table 5 showed the average time and standard deviation of the proposed pipeline. The average running time was 1.398 s with 2.473 fruits detected; i.e., successfully processing a fruit required 0.565 s on average. Therefore, the execution time of the developed method was reasonable for robotic harvesting. Euclidean clustering and branch reconstruction, which takes up most of the computational time, can be optimized to further improve the computational efficiency by (i) point cloud downsampling and (ii) making full use of the computational capability of the graphics processing unit.

## 5. Conclusions

Guava fruit detection and pose estimation are very important, as they can be used to guide a harvesting robot to approach the fruit without collisions with the branches, thus improving the harvest success rate. Therefore, this study investigated a vision sensing algorithm to detect the guava fruit and estimate its pose in real field conditions by using a low-cost RGB-D sensor, which comprised the following functions: (i) an FCN-based image segmentation method, (ii) a Euclidean clustering-based fruit detection method, (iii) a multiple 3D line segments detection method, and (iv) a pose estimator. The performance of the proposed algorithm was evaluated through experiments, and the following conclusions were summarized:(i)The FCN model realized a mean accuracy of 0.893 and an IOU of 0.806 for the fruit class, and obtained a mean accuracy of 0.594 and an IOU of 0.473 for the branch class. The result revealed that the guava fruit can be well segmented, but the branch was a little difficult to segment;(ii)The detection precision and recall of the proposed algorithm were 0.983 and 0.949, respectively. It can be concluded that the proposed algorithm was robust for detecting in-field guavas;(iii)The pose error of the bounding box-based method was $25.41{}^{\circ }\pm 14.73{}^{\circ }$, while that of the sphere fitting-based method was $23.43{}^{\circ }\pm 14.18{}^{\circ }$. The results suggested that the sphere fitting method was more suitable for pose estimation;(iv)The proposed pipeline needs 0.565 s on average to detect a fruit and estimate its pose, which was sufficient for a guava-harvesting robot.

In conclusion, the proposed vision sensing algorithm is able to detect guava fruits on trees and obtain promising 3D pose information with the use of a low-cost RGB-D sensor. Our future work will mainly focus on improving the success rate and precision of the 3D pose of the fruit.

## Figures and Tables

**Figure 1 sensors-19-00428-f001:**
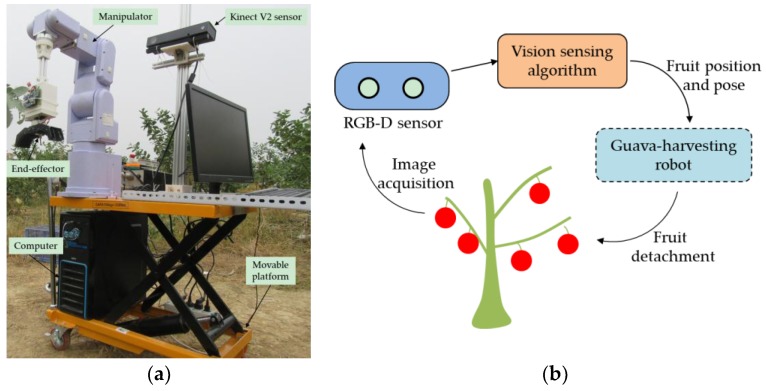
The guava-harvesting robot and its vision sensing system. (**a**) Robot system; (**b**) vision sensing system.

**Figure 2 sensors-19-00428-f002:**
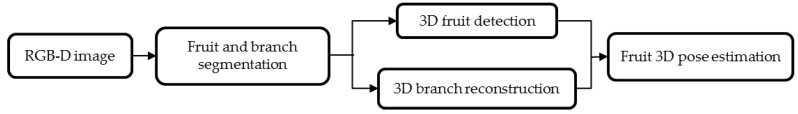
Flow diagram of the developed vision sensing algorithm.

**Figure 3 sensors-19-00428-f003:**
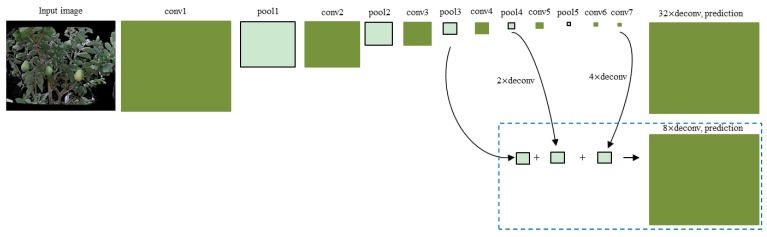
Fully convolutional network (FCN) configuration. The first row uses a deconvolution stride of 32, resulting in a coarse prediction. The second row fuses the outputs from the conv7 layer, the pool3 layer, and the pool4 layer at stride 8, leading to a finer prediction. The deconvolution parameter is defined as ‘(stride) × deconv’.

**Figure 4 sensors-19-00428-f004:**
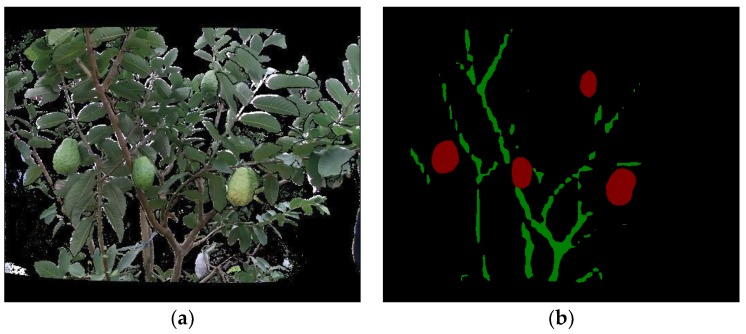
Segmentation results of the FCN model. (**a**) An aligned red–green–blue (RGB) image where black pixels represent objects outside the working range of the Kinect V2 sensor; (**b**) segmentation result where the red parts represent the fruits, and the green parts are the branches.

**Figure 5 sensors-19-00428-f005:**
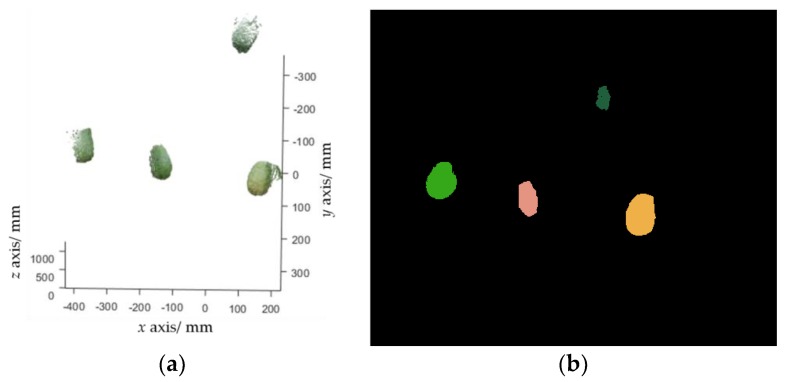
Fruit detection results. (**a**) Fruit point cloud extracted from Figure 4b; (**b**) clustering results, where each cluster is marked with a random color.

**Figure 6 sensors-19-00428-f006:**
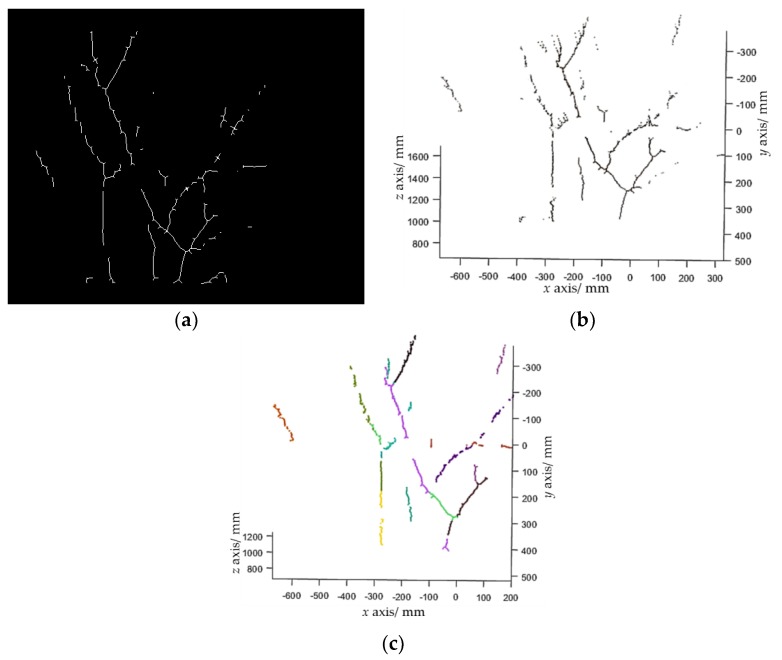
Branch reconstruction process. (**a**) Branch skeletons extracted from Figure 4b; (**b**) branch point cloud; (**c**) detected line segments, where each segment is marked with a random color.

**Figure 7 sensors-19-00428-f007:**
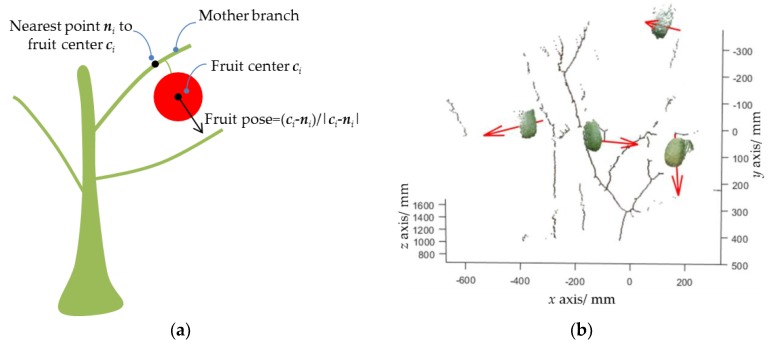
Principle of fruit pose estimation. (**a**) Schematic diagram; (**b**) three-dimensional (3D) pose estimation result, where the red array represents the fruit pose.

**Figure 8 sensors-19-00428-f008:**
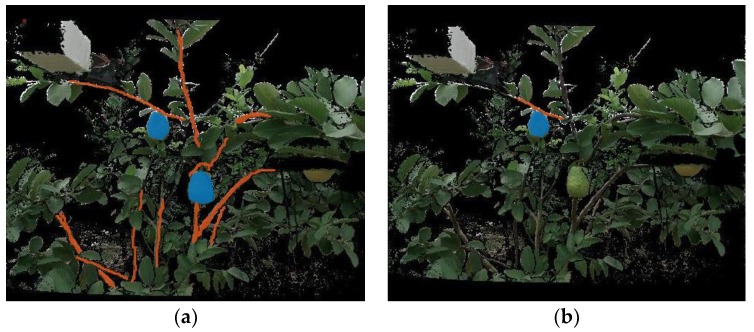
Example showing ground-truth labels. (**a**) Ground-truth labels for three classes: fruit (blue), branch (red), and background. (**b**) Ground-truth fruit (blue) and the corresponding mother branch (red).

**Figure 9 sensors-19-00428-f009:**
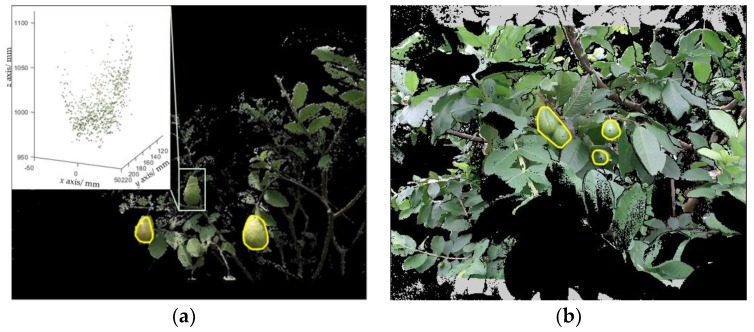
Examples illustrating unsuccessful detections.

**Figure 10 sensors-19-00428-f010:**
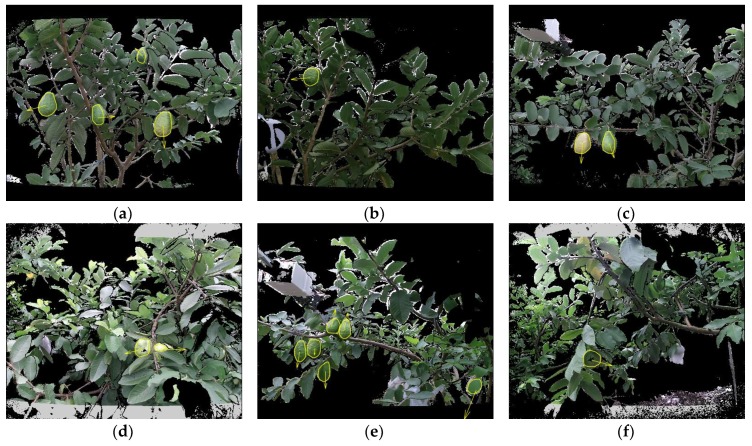
Examples illustrating the fruit poses estimated by the proposed algorithm. The yellow array represents the fruit pose.

**Figure 11 sensors-19-00428-f011:**
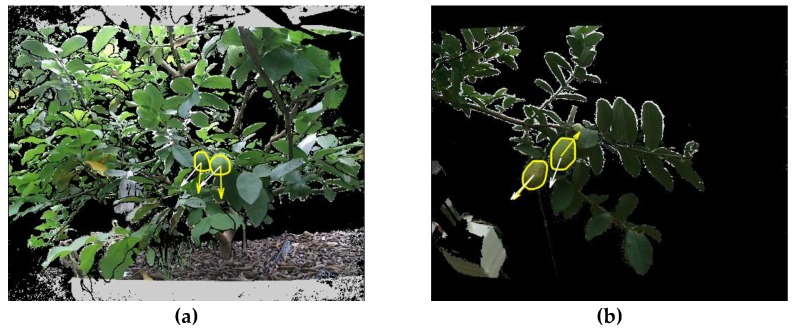
Failure examples. The yellow array represents the estimated pose, while the white array is the ground-truth pose.

**Table 1 sensors-19-00428-t001:** Mean accuracy and intersection over union (IOU) of the FCN, SegNet, and classification and regression trees classifier (CART) over the test set.

	Fruit		Branch	
	Mean Accuracy	IOU	Mean Accuracy	IOU
FCN	0.893	0.806	0.594	0.473
SegNet	0.818	0.665	0.642	0.389
CART	0.264	0.235	0.071	0.067

**Table 2 sensors-19-00428-t002:** Precision and recall of the proposed algorithm and the method in [4] over the test set.

Algorithm	# Images	# Fruits	# True Positives	# False Positives	Precision	Recall
Proposed	91	237	225	4	0.983	0.949
method in [4]	91	237	159	10	0.941	0.671

**Table 3 sensors-19-00428-t003:** Median error (MEDE) and median absolute deviation (MAD) of the 3D pose errors.

Method	MEDE (degree)	MAD (degree)
Bounding box	25.41	14.73
Sphere fitting	23.43	14.18

**Table 4 sensors-19-00428-t004:** Frequency of 3D pose error within certain angle limits.

Limit (Degree)	Bounding Box (%)	Sphere Fitting (%)
<45	70.45	74.24
<35	62.88	63.64
<25	49.24	53.79

**Table 5 sensors-19-00428-t005:** Real-time performance for the proposed algorithm over the test set.

Subtasks	Average Time (s)	Standard Deviation (s)
Segmentation	0.165	0.076
Fruit detection	0.689	0.368
Branch reconstruction	0.543	0.397
Pose estimation	0.000	0.000
Total	1.398	0.682

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
