# Peer review of "Guava Detection and Pose Estimation Using a Low-Cost RGB-D Sensor in the Field"

_sensors, 2019, doi:10.3390/s19020428_

Round 1
Reviewer 1 Report
This paper showed the fruit detection and pose estimation using convolutional network.
This results is better than previous reserach (Kusuman, 2017). However, i have the following coments.
1) I think that the number of data 200 is too small. Please change the combination of the learning data and the test data (Cross validation method), etc. and examine the generalization ability more.
2)I think whether the performance when Kinect V2 is used outside depends on the external environment(llumination). This paper used the images contained all kinds of illuminations.Did the error of depth and position change depending on the illumination? Please explain and/or discuss this point.
Author Response
The authors would like to thank the reviewer for the comments. The responses to these comments and the revisions implemented in the paper are detailed below. The reviewer' comments are in regular font, and our modification part is in blue, italic font.
Point 1: I think that the number of data 200 is too small. Please change the combination of the learning data and the test data (Cross validation method), etc. and examine the generalization ability more.
Response 1: Revised as suggested. We added 237 images in the dataset and used 80% of the images for training and 20% for testing as done by [10]. Ten-fold cross-validation was applied over the training set to determine an optimal FCN. The segmentation results of the FCN on the test set showed that the mean accuracy and IOU for the fruit class improved from 0.831 and 0.759 to 0.893 and 0.806, respectively, and those for the branch class improved from 0.541 and 0.410 to 0.594 and 0.473, respectively, after using more training data. The revised details can be found on Lines 170-171, 262, 263-265, and in Table 1:
In addition, ten-fold cross-validation was applied over the training set to determine an optimal FCN.
There were 437 RGB-D images captured in total.
The training set contained approximately 80% of the RGB-D images in the dataset and the test set included the remaining images as suggested by [10].
Table 1. Mean accuracy and IOU of FCN, SegNet and CART over the test set
Fruit | Branch | ||||
Mean accuracy | IOU | Mean accuracy | IOU | ||
FCN | 0.893 | 0.806 | 0.594 | 0.473 | |
SegNet | 0.818 | 0.665 | 0.642 | 0.389 | |
CART | 0.264 | 0.235 | 0.071 | 0.067 | |
[10] Sa, I.; Ge, Z.; Dayoub, F.; Upcroft, B.; Perez, T.; Mccool, C.. Deepfruits: a fruit detection system using deep neural networks. Sensors 2016, 16, 1222.
Point 2: I think whether the performance when Kinect V2 is used outside depends on the external environment (illumination). This paper used the images contained all kinds of illuminations. Did the error of depth and position change depending on the illumination? Please explain and/or discuss this point.
Response 2: Thanks for the reviewer’s advice. The depth precision is influenced by illumination changes. Specifically, the depth precision of Kinect V2 is depending on the light intensity, incidence angle and sensor-to-object distance [40]. On Lines 333-335 and Lines 337-338 in Section 4.2, the following sentences were added to explain the influence of illuminations.
(i) in strong sunlight, the standard deviation of the depth data of Kinect V2 for a sensor-to-target distance z = 1 m was up to 32 mm as reported by Fankhauser et al. [40], which would increase invalid points in the point cloud and result in detection failures (Figure 9a);
To address these two problems, the following solutions can be adopted: (i) avoiding using Kinect V2 at noon or using a light shield to block sunlight [15];
[15] Nguyen, T. T.; Vandevoorde, K.; Wouters, N.; Kayacan, E.; Baerdemaeker, J. G. D.; Saeys, W. Detection of red and bicoloured apples on tree with an rgb-d camera. Biosyst. Eng. 2016, 146, 33-44.
[40] Fankhauser, P.; Bloesch, M.; Rodriguez, D.; Kaestner, R.; Hutter, M.; Siegwart, R. Kinect v2 for Mobile Robot Navigation: Evaluation and Modeling. In Proceedings of the International Conference on Advanced Robotics (ICAR), Istanbul, Turkey, 27–31 July 2015; pp. 388–394.
Reviewer 2 Report
This paper presents a method to detect and locate (guava) fruit. The work is a precursor to robotic harvesting in context of gripper positioning to effectively grasp fruit from orientation/pose information, and is significant to the current literature. The technical details have been well described. The submission is clearly written in appropriate english (except for one lapse to first person, li 246)
I recommend the paper to be published after addressing the following comments.
Comments:
- An image is given of an assembly with a manipulator and end effector, but nothing is mentioned of this in text. Please expend - was the RGBD information used to guide the manipulator? What was the harvest success rate? What was the time to harvest? The likely response is that this will be reported into another manuscript, but my recommendation is for one very solid, very citeable paper over two least-publishable-unit papers.
- There is a flood of papers applying CNN to fruit detection/localisation beginning to appear, but there area is yet young, and new publications relevant. However, to make progress either new architectures should be tested on a common (public) dataset, or
- The relevance of 'poise' estimation to obstacle avoidance and the issue of obstacles 'unrelated' to the branch sub-tending the fruit (i.e. other branches) should be better explained
- Kinect V2 RGB-D cameras use 860 nm light for depth sensing, and sun light can interfere with the depth measurement. Were images acquired in sunlight or after sun set?
- There is mention of a 3D point cloud. My assumption is the X and Y axes on the 2D image plane and the Z axis from the depth map. Is it a stationary RGBD from a single viewpoint or is the point cloud created from capturing images from multiple viewpoints?
- In Eq (1), how were the camera principal points and focal lengths obtained?
- The input to FCN is not explained properly- is it just the RGB image or the RGB-D ? Has any resizing of the image occurred?- what was input size to FCN? In Fig 4(a), did the authors down-sample the Kinect RGB images to adapt the depth information or vice versa?
- Fruit positions in Fig 4(a,b) and Fig 5b are consistent while not in Fig 4(a). Please explain why.
- Assuming that FCN is able to segment and classify the fruits and branches from their background, why was an additional Euclidean cluster used to detect the fruit again? The Kinect depth information itself contains quite a lot of errors such as occlusion, mapping and distance estimation error. Besides, the leaves and branches touching the fruit can have very similar depth information such that the point cloud are difficult to differentiate fruit from its background. By contrast, the fruit has smooth and convex surface, and they are easier to be detected in RGB images. A comparison of fruit/branch detection accuracy between FCN and FCN+ Euclidean clustering is desirable.
- Line 164 - “point cloud of the fruit map of the FCN output”. Please clearly explain the output of the FCN. Is it a point cloud or the output feature map associated to the depth map from Kinect?
- Line 240 “training set contained 40% of RGBD images”. Why is it only 40% and not something commonly used in deep learning like train-test splits of (90-10)% or (80-20)%? FCN could have produced better results with more training data rather than just 80 training images.
- It was mentioned that the FCN was trained in Caffe. Please mention the web link to the FCN code repository and how was the training implemented? Does it involved fine-tuning and/or transfer learning?
- Line 280 should read “CART” instead of CATR.
- Please provide background on why the branch hanging the fruit is useful to guide a robot harvester? I would have thought that real-word coordinates of fruit centre relative to camera is more important to guide the gripper. Please provide an equation of how to calculate the angle between the estimated pose and the ground-truth pose, and explain further such as what’s the significance of 0 degree.
- As stated in line 309~315, the training set used in the article is insufficient, and the authors are encouraged to add more data set to improve their results. It is common to take a few weeks to label training set when using deep learning techniques. By contrast, the authors just spent 2 days to prepare their training set which is insufficient.
- Line 344, one of the cause of failures “mother branch of fruit was very thin or invisible” which could relate to line 189 as an impact of “thinning after skeleton extraction”?
li 365 restate computing hardware used in speed test
Wording issues:
- Line 23 drop the word challenging or reword the sentence.
- Figure 1 shows the robot arm picking the fruit. It would have been better if the practical in-field experiments of fruit picking were added especially to access the practicability with the current pose estimation error 23.35 degree +/- 15.87 degree in the current study.
- Line 344, one of the cause of failures “mother branch of fruit was very thin or invisible” which could relate to line 189 as an impact of “thinning after skeleton extraction”?
- Line 14, change “To conduce” to “To conduct”
Author Response
The authors would like to thank the reviewer for the comments. The responses to these comments and the revisions implemented in the paper are detailed below. The reviewer' comments are in regular font, and our modification part is in blue, italic font.
Point 1: An image is given of an assembly with a manipulator and end effector, but nothing is mentioned of this in text. Please expend - was the RGBD information used to guide the manipulator? What was the harvest success rate? What was the time to harvest? The likely response is that this will be reported into another manuscript, but my recommendation is for one very solid, very citeable paper over two least-publishable-unit papers.
Response 1: Thanks for the reviewer’s advice. We had conducted an in-field harvest experiment on September 15 and October 13, 2018, when the vision system provided only the fruit position information from RGB-D images. The harvest success rate was only 48.05% (37/77), and the average harvest time was 26.4 s. About 30% of the failures were due to collisions between the end-effector of the robot and the mother branch of the fruit. This experiment raised a question—how to calculate the fruit pose along which the end-effector can approach the fruit without collisions with the mother branch of the fruit? Therefore, this study investigated a fruit pose estimation algorithm. In short, the in-field experiment was carried out before this study, so its result was not discussed in this study.
Point 2: There is a flood of papers applying CNN to fruit detection/localisation beginning to appear, but there area is yet young, and new publications relevant. However, to make progress either new architectures should be tested on a common (public) dataset, or
Response 2: Thanks for the reviewer’s suggestion. FCN had been tested by Shelhamer et al. on a public dataset and showed impressive performance. So it was deployed in this study to segment RGB images. Our future work will develop a shallow FCN for embedded systems and will test it on public dataset as suggested.
Shelhamer, E.; Long, J.; Darrell, T. Fully convolutional networks for semantic segmentation. PAMI. 2017, 39(4), 640-651.
Point 3: The relevance of 'poise' estimation to obstacle avoidance and the issue of obstacles 'unrelated' to the branch sub-tending the fruit (i.e. other branches) should be better explained
Response 3: Revised as suggested. The first question was explained in Introduction (Lines 36-40):
if only the fruit position information is available, the end-effector of the harvesting robot is likely to have collisions with the mother branch of the fruit when moving toward a fruit, thus lowering the harvest success rate. Thus, for each fruit, estimating a three-dimensional (3D) pose relative to its mother branch along which the end-effector can approach the fruit without collisions is very important.
The second question was explained in Section 4.3 (Lines 363-365):
It is important to note that for branches without fruits, if they block the end-effector from approaching the fruit, the path planning algorithm proposed by our research group [41] can be utilized to avoid them.
[41] Cao, X.; Zou, X.; Jia, C.; Chen, M.; Zeng. Z. RRT-based path planning for an intelligent litchi-picking manipulator. Comput. Electron. Agric. 2019, 156, 105-118.
Point 4: Kinect V2 RGB-D cameras use 860 nm light for depth sensing, and sun light can interfere with the depth measurement. Were images acquired in sunlight or after sun set?
Response 4: Thanks for the reviewer’s comment. The images were acquired in sunlight from 8:00 a.m. to 5:00 p.m. Based on the experiment results, the noon sunlight had a large impact on the robustness of the developed algorithm, while the morning sunlight and afternoon sunlight had little effects on the algorithm. A possible explanation is that the noon sunlight contains a certain amount of light with a wavelength of 860 nm which interferes with the depth precision of Kinect V2 [40]. For clarity, the following sentences were added on Lines 333-335 and Lines 337-338 in Section 4.2 to explain the influence of sunlight:
(i) in strong sunlight, the standard deviation of the depth data of Kinect V2 for a sensor-to-target distance z = 1 m was up to 32 mm as reported by Fankhauser et al. [40], which would increase invalid points in the point cloud and result in detection failures (Figure 9a);
To address these two problems, the following solutions can be adopted: (i) avoiding using Kinect V2 at noon or using a light shield to block sunlight [15];
[15] Nguyen, T. T.; Vandevoorde, K.; Wouters, N.; Kayacan, E.; Baerdemaeker, J. G. D.; Saeys, W. Detection of red and bicoloured apples on tree with an rgb-d camera. Biosyst. Eng. 2016, 146, 33-44.
[40] Fankhauser, P.; Bloesch, M.; Rodriguez, D.; Kaestner, R.; Hutter, M.; Siegwart, R. Kinect v2 for Mobile Robot Navigation: Evaluation and Modeling. In Proceedings of the International Conference on Advanced Robotics (ICAR), Istanbul, Turkey, 27–31 July 2015; pp. 388–394.
Point 5: There is mention of a 3D point cloud. My assumption is the X and Y axes on the 2D image plane and the Z axis from the depth map. Is it a stationary RGBD from a single viewpoint or is the point cloud created from capturing images from multiple viewpoints?
Response 5: Thanks for the reviewer’s question. The RGB-D image was created from a single, stationary viewpoint. The resulting partial point cloud has been shown sufficient for fruit detection by Nguyen et al. For clarity, the following sentence was added on Lines 141-142 in Section 2.2:
Note that each point cloud is created from a single viewpoint, so it only contains part of the geometry of the object. Nevertheless, partial point clouds are sufficient for fruit detection [15].
[15] Nguyen, T. T.; Vandevoorde, K.; Wouters, N.; Kayacan, E.; Baerdemaeker, J. G. D.; Saeys, W. Detection of red and bicoloured apples on tree with an rgb-d camera. Biosyst. Eng. 2016, 146, 33-44.
Point 6: In Eq (1), how were the camera principal points and focal lengths obtained?
Response 6: Revised as suggested. The calculation of camera intrinsic parameters was explained on Lines 127-128 in Section 2.1:
Ux, Uy, fx and fy were estimated using the calibration method developed by Zhang [30].
[30] Zhang, Z. A flexible new technique for camera calibration. PAMI. 2002, 22(11), 1330-1334.
Point 7: The input to FCN is not explained properly- is it just the RGB image or the RGB-D ? Has any resizing of the image occurred?- what was input size to FCN? In Fig 4(a), did the authors down-sample the Kinect RGB images to adapt the depth information or vice versa?
Response 7: Thanks for the reviewer’s question. The input to FCN was just the RGB image. All RGB images including Fig. 4a were resized to 424×512 pixels to match the depth image as mentioned in Section 2.1. Therefore, the input size to FCN was 424×512 pixels. The following sentence was added to explain the FCN input on Line 165 in Section 2.2.1:
Here, the input to FCN is the aligned RGB image with a resolution of 424×512 pixels.
Point 8: Fruit positions in Fig 4(a,b) and Fig 5b are consistent while not in Fig 4(a). Please explain why.
Response 8: Thanks for the reviewer’s comment. Fig. 5a was a point cloud that was showed from a viewer angle. This viewer angle caused the inconsistency. For clarity, a coordinate system was added to emphasize that Fig. 5a was a point cloud extracted from Fig. 4b.
Point 9: Assuming that FCN is able to segment and classify the fruits and branches from their background, why was an additional Euclidean cluster used to detect the fruit again? The Kinect depth information itself contains quite a lot of errors such as occlusion, mapping and distance estimation error. Besides, the leaves and branches touching the fruit can have very similar depth information such that the point cloud are difficult to differentiate fruit from its background. By contrast, the fruit has smooth and convex surface, and they are easier to be detected in RGB images. A comparison of fruit/branch detection accuracy between FCN and FCN+ Euclidean clustering is desirable.
Response 9: Thanks for the reviewer’s advice. Euclidean clustering was a necessary step in the developed pipeline for two reasons: (1) FCN was unable to identify how many fruits were in the fruit binary image it outputted, so Euclidean clustering was used to recognize each individual fruit from the FCN output; and (2) Because FCN may segment adjacent fruits into a single region, Euclidean clustering was utilized to split such a region into multiple parts. Figure A1 shows the necessity of Euclidean clustering. For clarity, the following sentences were added on Lines 180-182 in Section 2.2.2 to explain the necessity of Euclidean clustering:
Because FCN is unable to identify how many fruits are in the fruit binary map it outputs and may segment adjacent fruits into a single region, it is necessary to extract all individual fruits from the FCN output to realize fruit detection.
Note: Figure A1 can be found in the attachment.
Point 10: Line 164 - “point cloud of the fruit map of the FCN output”. Please clearly explain the output of the FCN. Is it a point cloud or the output feature map associated to the depth map from Kinect?
Response 10: Thanks for the reviewer’s question. The FCN outputted a fruit and a branch binary map from an RGB image which has been resized to match the depth map. The explanation of the FCN output was added on Lines 171-172 in Section 2.2.1 and Lines 183-184 in Section 2.2.2:
After training, FCN can be applied to segment the aligned RGB image to output a fruit and a branch binary map.
Let pfruit denote a set of pixels that belong to the fruit class in the fruit binary map outputted by FCN. pfruit can be transformed into a point cloud, defined as Pfruit, by using Eq. 1.
Point 11: Line 240 “training set contained 40% of RGBD images”. Why is it only 40% and not something commonly used in deep learning like train-test splits of (90-10)% or (80-20)%? FCN could have produced better results with more training data rather than just 80 training images.
Response 11: Thanks for the reviewer’s suggestion. Our previous experience on machine learning was used to split the dataset. But in deep learning, such a split cannot provide enough training data for learning. Therefore, we added 237 images in the dataset and used 80% of the images for training and 20% for testing as done by [10] to produce a better FCN. The segmentation results of FCN showed that the mean accuracy and IOU for the fruit class improved from 0.831 and 0.759 to 0.893 and 0.806, respectively, and those for the branch class improved from 0.541 and 0.410 to 0.594 and 0.473, respectively, after using more training data. The revised details can be found on Lines 262, 263-265 and in Table 1:
There were 437 RGB-D images captured in total.
The training set contained approximately 80% of the RGB-D images in the dataset and the test set included the remaining images as suggested by [10].
Table 1. Mean accuracy and IOU of FCN, SegNet and CART over the test set
Fruit | Branch | ||||
Mean accuracy | IOU | Mean accuracy | IOU | ||
FCN | 0.893 | 0.806 | 0.594 | 0.473 | |
SegNet | 0.818 | 0.665 | 0.642 | 0.389 | |
CART | 0.264 | 0.235 | 0.071 | 0.067 | |
[10] Sa, I.; Ge, Z.; Dayoub, F.; Upcroft, B.; Perez, T.; Mccool, C.. Deepfruits: a fruit detection system using deep neural networks. Sensors 2016, 16, 1222.
Point 12: It was mentioned that the FCN was trained in Caffe. Please mention the web link to the FCN code repository and how was the training implemented? Does it involved fine-tuning and/or transfer learning?
Response 12: Revised as suggested. The web link to the FCN code was mentioned on Lines 288-289 in Section 4:
All codes were programmed in MATLAB 2017b, except the FCN, which was implemented in Caffe [38] using a publicly available code [33].
[33] GitHub. FCN, 2018. https://github.com/shelhamer/fcn.berkeleyvision.org.
Fine-tuning was used to train the FCN, which was detailed in Section 2.2.1 in the original manuscript.
Point 13: Line 280 should read “CART” instead of CATR.
Response 13: Revised as suggested.
Point 14: Please provide background on why the branch hanging the fruit is useful to guide a robot harvester? I would have thought that real-word coordinates of fruit centre relative to camera is more important to guide the gripper. Please provide an equation of how to calculate the angle between the estimated pose and the ground-truth pose, and explain further such as what’s the significance of 0 degree.
Response 14: Thanks for the reviewer’s comments. The coordinates of fruit centre may be insufficient to guide the end-effector to approach the fruit without collisions. A visual proof was shown in Figure A2. Besides, Bac et al. has shown that the grasp success rate increased from 41% to 61%, when a stem-dependent pose of the fruit was considered. The background on the importance of the mother branch of the fruit was added on Lines 36-43 in Section 1:
if only the fruit position information is available, the end-effector of the harvesting robot is likely to have collisions with the mother branch of the fruit when moving toward a fruit, thus lowering the harvest success rate. Thus, for each fruit, estimating a three-dimensional (3D) pose relative to its mother branch along which the end-effector can approach the fruit without collisions is very important. In this work, the fruit pose is defined as a vector that passes through the fruit center and is perpendicular to the mother branch of the fruit. Bac et al. has shown that such a pose could increase the grasp success rate from 41% to 61% [3].
[3] Bac, C. W.; Hemming, J.; Van Tuijl, J.; Barth, R.; Wais, E.; Van Henten, E. J. Performance evaluation of a harvesting robot for sweet pepper. J. Field Robotics. 2017, 36, 1123-1139.
An equation that shows how to calculate the angle between the estimated and the ground-truth poses was added on Lines 348-349 in Section 4.3.
The significance of small angles was explained on Line 350 in Section 4.3 as follows:
Smaller angles correspond to smaller errors, and higher angles correspond to higher errors.
Note: Figure A2 can be found in the attachment.
Point 15: As stated in line 309~315, the training set used in the article is insufficient, and the authors are encouraged to add more data set to improve their results. It is common to take a few weeks to label training set when using deep learning techniques. By contrast, the authors just spent 2 days to prepare their training set which is insufficient.
Response 15: Revised as suggested. We added images in the dataset and redone all experiments. The revised details can be found on Line 262 and in Table 2-4:
There were 437 RGB-D images captured in total.
Table 2. Precision and recall of the proposed algorithm and the method in [4] over the test set
Algorithm | # images | # fruits | # true positives | # false positives | Precision | Recall |
Proposed | 91 | 237 | 225 | 4 | 0.983 | 0.949 |
method in [4] | 91 | 237 | 159 | 10 | 0.941 | 0.671 |
Table 3. Median and median absolute deviation of the 3D pose errors
Method | MEDE (degree) | MAD (degree) |
Bounding box | 25.41 | 14.73 |
Sphere fitting | 23.43 | 14.18 |
Table 4. Frequency of 3D pose error within certain angle limits
Limit ( degree) | Bounding box (%) | Sphere fitting (%) |
< 45 | 70.45 | 74.24 |
< 35 | 62.88 | 63.64 |
< 25 | 49.24 | 53.79 |
Point 16: Line 344, one of the cause of failures “mother branch of fruit was very thin or invisible” which could relate to line 189 as an impact of “thinning after skeleton extraction”?
Response 16: Thanks for the reviewer’s advice. We did not express ourselves clearly here. On the one hand, when the branches were very thin or invisible, the FCN was unlikely to recognize them. Consequently, the fruit pose relative to the branch cannot be estimated. On the other hand, the branches retained their original shapes after skeleton extraction, so they can be still reconstructed and used to estimate the fruit pose. In short, there was little association between Line 344 and Line 189. For clarity, the following sentence was added on Lines 372-374 in Section 4.3:
(i) when the mother branch of a fruit was very thin or invisible, the FCN was unlikely to recognize it, and hence the pose of this fruit relative to its mother branch could not be estimated.
Point 17: li 365 restate computing hardware used in speed test
Response 17: Revised as suggested.
The real-time experiment was implemented on a computer running a 64-bit Windows 10 system with 16 GB RAM, NVIDIA GeForce GTX 1060 6GB GPU, and Intel core i5-3210M CPU.
Point 18: Line 23 drop the word challenging or reword the sentence.
Response 18: Revised as suggested.
Point 19: Figure 1 shows the robot arm picking the fruit. It would have been better if the practical in-field experiments of fruit picking were added especially to access the practicability with the current pose estimation error 23.35 degree +/- 15.87 degree in the current study.
Response 19: Thanks for the reviewer’s suggestion. The carmine guava in Guangzhou, China, matures from July to November. So it may not be the time to conduct a field test. In addition, after an in-field experiment on September 15 and October 13, 2018, we found three tough problems to be addressed: (1) the robot had collisions with the mother branch of the fruit (the research topic of this study); (2) the inverse kinematics solver of the manipulator was a little unstable; and (3) the end-effector may cause bruises on the surface of the fruit. Our future work will solve problems 2 and 3 to optimize the robot and then conduct field tests to verify the practicability of the developed algorithm.
Point 20: Line 344, one of the cause of failures “mother branch of fruit was very thin or invisible” which could relate to line 189 as an impact of “thinning after skeleton extraction”?
Response 20: See response 16.
Point 21: Line 14, change “To conduce” to “To conduct”
Response 21: Revised as suggested.
In addition to the above changes, we have also modified the sentence containing the first person.

Reviewer 3 Report
The current manuscript demonstrates an interesting harvesting approach based on fruit detection and pose detection of guava crops through FCN, SegNet and CATR models. The presented method is applicable to real field conditions and capable of achieving precision and recall of guava fruit detection equal to 0.964 and 0.935 respectively. Regarding pose estimation, sphere fitting and bounding box approaches were employed, concluding that the first has proven more trustworthy and accurate (the pose error has been estimated between 23.35° ± 15.87°). The presented approach contributes indeed to crops automatic harvesting since it can be embedded to agricultural robotic systems. The manuscript is encouraged for further publication. However, there are some recommendations that authors are suggested to take into consideration:
Abstract
Lines 13-16. Please reform the two sentences in a way to make clearer to the reader the aim, the importance and the main reason that lead to the presented study.
Lines 18-22. Please avoid numbering the utilized steps in the abstract section. The authors are suggested to provide a very short reference of the four steps that were followed in the current approach.
Introduction
Lines 56-57 Why RGB-D images are considered more informative ? The authors are suggested to elaborate more on this area by providing relevant scientific reference.
Lines 67-68 The authors are suggested to elaborate more.
Line 111. In Equation 1 it is not mentioned what does zi denote.
Author Response
The authors would like to thank the reviewer for the comments. The responses to these comments and the revisions implemented in the paper are detailed below. The reviewer' comments are in regular font, and our modification part is in blue, italic font.
Point 1: Lines 13-16. Please reform the two sentences in a way to make clearer to the reader the aim, the importance and the main reason that lead to the presented study.
Response 1: Revised as suggested. The revised details can be found on Lines 13-16:
Fruit detection in real, outdoor conditions is necessary for automatic guava harvesting, and the branch-dependent pose of fruits is also crucial to guide a harvesting robot to approach and detach the target fruit without collisions with its mother branch. To conduct automatic, collision-free picking, this study investigates a fruit detection and pose estimation method by using a low-cost RGB-D sensor.
Point 2: Lines 18-22. Please avoid numbering the utilized steps in the abstract section. The authors are suggested to provide a very short reference of the four steps that were followed in the current approach.
Response 2: Thanks for the reviewer’s advice. The sentences on Lines 17-22 were rephrased as suggested. References for the developed pipeline were added on Lines 106-110 as suggested.
Point 3: Lines 56-57 Why RGB-D images are considered more informative? The authors are suggested to elaborate more on this area by providing relevant scientific reference.
Response 3: Revised as suggested. The revised details can be found on Lines 60-63:
As RGB-D images encode the color and 3D geometry of the object, and the RGB-D depth image is invariant to illumination changes, RGB-D images are more informative than RGB images. Therefore, there has been an increase in using RGB-D sensors to detect fruits [4].
Point 4: Lines 67-68 The authors are suggested to elaborate more.
Response 4: Revised as suggested. The revised details can be found on Lines 73-75:
In conclusion, both the CNN-based and RGB-D-based methods show promising results on fruit detection in the fields. This study fuses these two methods to detect guava fruits in outdoor conditions.
Point 5: Line 111. In Equation 1 it is not mentioned what does zi denote.
Response 5: Revised as suggested. The revised details were given as follows:
are the 3D coordinates of pixel i.